# Applying a Goal-Directed Behavior Model to Determine Risk Perception of COVID-19 and War on Potential Travelers’ Behavioral Intentions

**DOI:** 10.3390/ijerph20032562

**Published:** 2023-01-31

**Authors:** Taeuk Kim, Jungwoo Ha

**Affiliations:** 1Department of Hotel & Restaurant Management, Kyonggi University, Seoul 03746, Republic of Korea; 2Department of Tourism Event Management, Kyonggi University, Seoul 03746, Republic of Korea

**Keywords:** COVID-19 risk perception, war risk perception, uncertainty toward international travel, mental well-being toward international travel, desire, behavioral intentions toward international travel

## Abstract

The purpose of this study is to verify the influence of the relationship between risk perception of COVID-19 and the war-applied Model of Goal-directed Behavior (MGB) based on stimulus–organism–response (SOR) and potential travelers’ behavioral intention. In addition, this study attempted to verify the relationship among uncertainty toward international travel, mental well-being toward international travel, and desire toward travelers’ behavioral intention. Moreover, we examined the moderating effect of gender (female vs. male) among all variables for dependents. The survey was conducted on potential travelers in Korea. As for the survey period, a survey was conducted for one month beginning on 2 September 2022. Of the total 413 surveys, 361 surveys were used for the final analysis, and 52 unfaithful surveys were excluded. In addition, demographic, CFA, correlation analysis, structural equation modeling, and moderation effect analysis were verified using SPSS and AMOS. For the data analysis, we used SPSS 18.0 and Amos 20.0 to perform factor analysis and SEM. Significant effects were found in support for Hypotheses 1–5. Further, when it comes to the difference of gender on the relationship between all the variables, while no significant effect was found for Hypotheses 6a,c,e,g, a significant effect was found for Hypotheses 6b,d,f. Thus, H6a,c,e were rejected and H6b,d,f were supported. It was found that females had a greater influence on mental health and desire for overseas travel than males, but it was found that there was no difference between females and males in the relationship between desire and behavioral intention. Therefore, it was possible to verify that the MGB desire is an important psychological variable for both females and males. Furthermore, these findings offer academic practical implications to travel and tourism companies by presenting basic data based on the results of empirical research analysis in the context of the current dangerous situation.

## 1. Introduction

At the end of 2021, expectations of overcoming the COVID-19 pandemic and recovering international tourism were delayed following a surge in Omicron variant COVID-19 cases. As of 15 April 2022, the cumulative number of COVID-19 confirmed cases worldwide was 502,089,027; however, it is anticipated that international tourism will be revived, since the number of new confirmed cases has gradually decreased in the second quarter of 2022 [1]. The risks of past diseases such as Swine Flu and MERS have dampened the global tourism market and had a local and short-term impact while the COVID-19 crisis is an unprecedented risk in which the world has been paralyzed for more than a year, and international travel demand has almost been extinguished [2]. Scholars conducting research on the perceived risk of COVID-19 changing the world have discovered that it has caused a shift in travel decision-making behaviors [3,4].

With the recovery of daily life post-COVID-19 taking place mainly in the United States and Europe, efforts have begun to ease self-quarantine norms for inbound and outbound travelers to revive international tourism. However, international tourism is again diminishing in response to the Russia’s military attack on Ukraine in February 2022 [5]. The sudden outbreak of war has adversely affected the international community in various respects [6,7]. Specifically, prolonged war coupled with continuing concerns about COVID-19 could result in the loss of international tourism worth an estimated 14 billion dollars worldwide [8]. It is still a little early to measure the overall impact of the war on international tourism, but it is true that the resulting deterioration of the international economy may adversely affect international tourism demand in the longer term. Accordingly, it is necessary to examine the factors that will affect international tourism demand.

Potential travelers experience increased uncertainty about travel due to anxiety and fear of both COVID-19 and the war, which have a negative impact on their mental health [9]. Previous studies on COVID-19 risk perception among potential travelers [10,11,12,13,14] state that COVID-19 risk perception leads to negative behavior that results in hesitation concerning either the planning of or actual travel. Therefore, previous research has focused on identifying various types of travel-related risks [15]. Initially, Roehl and Fesenmaier [16] classified travel groups based on risk related to physical equipment, vacation, and destination. Reisinger and Mavondo [17] investigated the influence of risk associated with sociocultural, health financial, travel intention, and terrorism risks. Perko et al. [18] classified backpackers’ perceived risk into five categories: environmental, political, financial, socio-psychological, physical, and expectation. Among the various types of perceived risk, health risk refers to tourists’ or hospitality customers’ perceived risk to their physical health as a result of uncontrollable events such as terrorism, political unrest, natural disasters, and pandemics [19]. A growing body of research on this issue emerged in response to a series of events, such as the 9/11 terrorist attacks in 2001, the SARS outbreak in 2003, the Bali bombings in 2002, and the Asian tsunami in 2004 [15].

Against this background, various theories have been applied to examine the causal relationship between travelers’ desire for travel and behavioral intentions; particularly, the Model of Goal-directed Behavior, which is known to be effective in predicting travelers’ behavioral intentions [20]. According to the model, an individual acts with a certain purpose or goal, which provides the strongest indicator to predict their behavioral intentions and actual behavior [21,22,23]. However, the need for specific verification of risk in relation to COVID-19 and other threatening factors is raised as a variable that may negatively affect travelers’ behavioral intentions based on potential goals. Moreover, if the relationship between uncertain feelings toward international travel and psychological conditions such as mental well-being is verified, potential travelers will be able to understand their decision-making process more clearly in dangerous situations. Moreover, additionally applying these variables to the Model of Goal-directed Behavior, we aim to verify the structural relationship with travelers’ behavioral intentions.

Therefore, in this study, hypotheses are established based on existing studies [22,23,24] that have been validated as a useful theory for predicting travelers’ behavior, and the psychological conditions of potential travelers facing the current uncertain situation are further verified. Thus, the specific purpose of this study is as follows. First, we examine the relationship between risk perception of COVID-19, uncertainty about international travel, and mental well-being; second, we examine the relationship between risk perception of war, uncertainty about international travel, and mental well-being; third, we examine the relationship between uncertainty about international travel, mental well-being, and potential travelers’ desire; fourth, we examine the relationship between desire and travel intention. In addition, the moderation effect was conducted to identify differences according to gender. Based on the results of this study, we will provide useful basic information for formulating future marketing strategies in the tourism industry, which has suffered stagnation since 2020.

## 2. Theoretical Background

### 2.1. COVID-19/War Risk Perception

Risk perception refers to self-awareness of the possibility that one’s actions can have negative consequences [25], which is an important variable in understanding and predicting consumer behavior. Risk perception can occur differently depending on the type of purchase or seller from a contextual perspective. Even under the same circumstances, the level of risk perception may vary depending on an individual’s propensity for and degree of risk acceptance. In other words, those who have a large capacity for risk acceptance are willing to accept risk, but those with a reduced capacity strive to reduce risk perception [26].

Analysis and management of risk perception are very important in highly involved and intangible service-based products such as travel. In particular, risk perception of health problems in tourism can be a very sensitive matter, whereby increased perceived travel risk negatively affects potential tourists’ intention to visit destinations. These consumer behaviors tend to be greatly influenced by psychological risk perception rather than actual experienced risk.

International travel, which is characterized by a lack of information, is more likely to trigger perceptions of psychological risk than domestic travel. Such perceptions have a significant influence on tourist behavioral intentions [27]. In other words, risk perception refers to awareness of the possibility of occurrence and degree to which tourists experience risk factors (such as disasters, disease, fraud, and crimes) that are subjectively unpredictable within tourist destinations [28]. Modern tourism risk factors vary widely and include economic crisis, terrorism, natural disasters caused by climate change, and disease that can spread directly; moreover, these are not limited to specific regions because of exchanges between countries [29].

In the hospitality industry, a number of studies have considered the risk perception of travelers relating to COVID-19. A further danger perceived by travelers is terrorism as a threat of war. Certain destinations are targeted by terrorists because they are convenient to access and can provide varied opportunities for terrorist actions. Transportation facilities, especially international airports, provide places in which terrorists can act because many foreigners come and go, making it easy to bring in weapons and to escape capture [30]. In this context, travelers who feel a sense of threat and anxiety about a destination will have a negative perception of that destination, and this effect can pose a significant threat to the tourism industry. First, potential travelers will decide to avoid visiting their destination because of a reputation of a high crime rate; second, if they feel insecure in their target destination, they will refrain from taking part in outdoor activities; and third, if they feel threatened or insecure, they will not revisit the destination and not recommend it to others [31].

Risk perception related to tourism is classified as temporal, satisfactory, psychological, social, physiological, safety, and capital [4]. In previous studies related to disease and tourism behavior, COVID-19 was identified as a psychological risk by [32], an emotional risk and cognitive risk by [33], and a perceived acceptability and perceived severity risk by [34]. In this study, we use risk perception of COVID-19 as a variable in the model by defining and organizing it as a single variable of COVID-19 disease risk related to travel. In addition, the risk perception of war is modified and supplemented to suit the purpose of this study as a single variable as used in the study by Kim, Choi, and Leopkey [35]. Accordingly, this study examines the psychological state and behavior of travelers according to risk perception (COVID-19/war) in terms of stimulus–organism–response (SOR) as mentioned by Mehrabian and Russell [36]. SOR is a framework that has been widely used to verify consumer responses to environmental impacts [37] and is a useful concept for examining the influence of emotional responses to external stimuli on behavior.

### 2.2. Model of Goal-Directed Behavior

When predicting and explaining personal behavior in the field of social science, the planned behavior theory proposed by Ajzen [38] is generally applied, although Perugini and Bagozzi [24] point out that it can only explain behavior from a cognitive perspective and does not include emotional factors that affect performance. The Theory of Planned Behavior is limited in that existing variables—such as attitude, subjective norms, and perceived behavior control—are cognitive factors that present justification for the behavior itself, and do not include clear motivational content for the behavior because they overlook the emotional aspect.

Perugini and Bagozzi [24] proposed a Model of Goal-directed Behavior to complement this and tried to grasp human behavioral intention through anticipated emotion and desire, including attitudes and subjective norms, which are constructs addressed in the existing planned behavior theory. The Model of Goal-directed Behavior further elaborates on the existing planned behavioral theory by adding the parameter of desire to induce motivation. Due to its outstanding predictive power, it has recently attracted attention as a means of understanding various human behaviors [39]. The Model of Goal-directed Behavior is used in a number of tourism fields due to its ease in expanding and supplementing the insufficient explanatory power of either the TRA or TPB.

Previous studies on the tourism industry that have applied the goal-oriented model are as follows. In Novani’s [40] study of travelers who experienced staycations in Indonesia, the risk perception of COVID-19 and non-pharmaceutical intervention, which affect the relationship between motivation, desire, and behavioral intention, were added as explanatory variables affecting the framework of the existing target-oriented model. As a result, it was found that desire had an effect on behavioral intention, and the risk perception of COVID-19 had an effect on aspiration, although non-pharmaceutical intervention had no influencing relationship. Das and Tiwari’s [41] study of international/domestic travelers in India surveyed travelers’ travel intentions during COVID-19. In the framework of the existing goal-oriented model, perceived severity of COVID-19 and personal non-pharmacological interventions were added, and all newly added variables were found to have a significant influence on desire and behavioral intention. In addition, demographic characteristics were added, showing that women and older travelers perceived the severity of COVID-19 to be higher, thereby embracing more personal non-pharmacological interventions to travel during COVID-19. These findings correspond with previous studies that include the risk perception of COVID-19 in the Model of Goal-directed Behavior, and previous studies on risk perception of war or similar environments related to destination presented in this study, as follows.

In a study of travelers from around the world visiting Hong Kong during the protest period, Kim et al. [42] found that risk perception constituted four sub-factors: physical risk, financial risk, privacy risk, and performance risk. Moreover, it was found that risk perception for all of these factors was influenced by desire and behavioral intention according to the Model of Goal-directed Behavior. A study by Larsen et al. [43] on international travelers to Mallorca, Spain, analyzed the psychological condition of people in relation to events such as the Iraq War (2003), the Spanish terrorist attack (2004), and the London bombing (2005), and found that the desire to travel had no significant effect despite people’s perception of risk in war or terrorist areas. In other words, the perception of anticipated negative emotions during travel (that is, expecting destinations to be dangerous) did not inhibit people from experiencing their desire to travel.

The concurrent risk perception of COVID-19 and war is the explanatory variable presented in this study. It is necessary to expand and examine from the risk perception of COVID-19 suggested in many previous studies. Accordingly, this study intends to examine the psychological situations and behaviors perceived by travelers, at a complex time when the risk of COVID-19 and war occurs simultaneously, by referring to the previous study and applying the Model of Goal-directed Behavior.

### 2.3. COVID-19 and War Risk Perception Link to Uncertainty and Mental Well-Being in Relation to International Travel

Cross-sectional and longitudinal data are available from the PsyCorona Survey (an international project on COVID-19) that included over 60,000 participants from 112 countries. As a result of analysis of the data gathered using a 20-min web-based survey, it was found that a high-risk perception of COVID-19 also had a significant negative effect on emotion and mental well-being [44].

The key distinction between perceived risk and perceived uncertainty remains unresolved. While some research considers risk and uncertainty to be the same concept [45], another line of research contends that risk and uncertainty are distinct psychological states [46]. Another study by Shi and Kim [47] on young adults in Singapore integrated the Risk Perception Attitude Framework and the Theory of Planned Behavior. They confirmed the relationship between people’s psychological factors and behaviors. The Theory of Planned Behavior and the Risk Perception Attitude Framework can complement each other to better understand the roles of psychosocial factors underlying mental health promotion behaviors. A study by Yan et al. [48] of 76 hotel employees in Peru applied the Transaction Theory of Stress and capping, showing that depressive symptoms increased during the COVID-19 pandemic, resulting in lower job satisfaction, particularly among employees with children, for whom the results were more severe. It has been confirmed in a number of previous studies that people with a higher risk perception for COVID-19 have adverse emotional and mental health. In a study by Chua et al. [34] targeting tourists in the United States, it was found that health risk perception had a significant positive (+) effect on mental well-being and perceived uncertainty. It was also confirmed that mental well-being had a statistically significant negative (−) effect on attitudes toward international travel, but perceived uncertainty did not influence the effect on attitude toward international travel.

Conversely, a study by Lee and Lemyre [49] on citizens in Canada confirmed that terrorism had a negative effect on psychological factors such as anxiety and fear. In addition, in a study by Kim et al. [35] targeting citizens based in the United States assuming South Korea as a travel destination, risk perception of terrorism was found to have a statistically negative influence on travel intention. Lastly, Fuchs and Reichel’s [50] study of international tourists visiting Israel analyzed the relationship between destination risk perception and its sub-factors—physical risk, human-induced risk, and socio-psychological risk—most of which showed a psychological correlation with negative factors such as fear and anxiety. As a result of the findings of the literature review, the following is hypothesized:

**Hypothesis 1a (H1a).** 
*COVID-19 risk perception has a significant effect on uncertainty concerning international travel.*


**Hypothesis 1b (H1b).** 
*COVID-19 risk perception has a significant effect on mental well-being concerning international travel.*


**Hypothesis 2a (H2a).** 
*War risk perception has a significant effect on uncertainty concerning international travel.*


**Hypothesis 2b (H2b).** 
*War risk perception has a significant effect on mental well-being concerning international travel.*


### 2.4. Uncertainty and Mental Well-Being Concerning International Travel Linked to Desire

The MGB contains positive/negative anticipated emotions for the success/failure of a goal, and these anticipated emotions are identified as “pre-factuals” that affect intention and behavior [51]. Several researchers have suggested that performance on behavior or the emotional response to non-performance can be an important determinant of behavioral intention [52]. In uncertain situations, people can easily feel either positive or negative emotions about future behavior, and negative anticipated emotions associated with failure to achieve goals play a role in predicting desire [24], which can eventually lead to positive/negative goals. Thus, positive/negative anticipated emotions serve to predict desire in the MGB.

In this respect, since a number of studies have confirmed that positive anticipated emotion has a positive influence on desire and negative anticipated emotion has a negative influence on desire in the Model of Goal-directed Behavior, we attempt in this study to verify the correlation with mental well-being as one of the psychological conditions. In addition, health-conscious people try to pursue their goals through various methods during travel. Thus, positive/negative anticipated emotion can infer the results of travelers’ low desire with high uncertainty and low mental well-being in relation to international travel, as shown by previous studies. For example, a study by Zhang et al. [53] of travelers who experienced rural tourism/ecotourism in China applied the Model of Goal-directed Behavior to show that travelers with a high degree of health consciousness and anticipated place emotion had a significant positive (+) effect on health-directed desire, and this desire has a significant effect on their behavioral intention to visit the rural eco-site. This helps to understand the psychological condition of travelers who want to pursue health issues; specifically, it can be seen that the positive psychological condition increases desire and contributes to increasing behavioral intention. Based on these studies, the following hypotheses were developed:

**Hypothesis 3 (H3).** 
*Uncertainty in relation to international travel has a significant effect on desire.*


**Hypothesis 4 (H4).** 
*Mental well-being in relation to international travel has a significant effect on desire.*


### 2.5. Desire Linked to Behavioral Intentions in Relation to International Travel

Perceived risks play a key role in the decision-making process of tourists [54]. People with a desire or intention to travel in a virus-infested situation take an important attitude toward health and safety management before going on their journey [55]. Desire is a powerful emotional state for subjects who act with psychological motivation [56]. Desire is the most central determinant in the Model of Goal-directed Behavior and is the most distinctive factor from the Theory of Planned Behavior consisting only of cognitive factors [24]. In addition, desire is an intermediary role that converts the correlation between attitude, subjective norms, and perceived behavioral control into motivation and directly affects behavioral intention. In other words, desire is a major predictor of behavioral intention, and plays a role in further increasing behavioral intention by mediating attitudes, subjective norms, expected emotions, and perceived behavioral control [24].

Previous studies in the tourism industry that relate to these are as follows. In a study related to perceived risk associated with Airbnb, Yi et al. [56] verified that attitudes, subjective norms, perceived behavioral control, and positive and negative expected emotions have a significant positive (+) effect on desire, and desire on behavioral intention. Song et al. [22] studied the decision-making process for participating in the oriental medicine festival using the MGB and found that attitudes, subjective norms, positive expectations, and frequency of past experience had a positive (+) effect on desire, and desire also had a positive (+) effect on behavioral intention. In addition, a study by Kement et al. [57] on Turkish travelers verified the influence of COVID-19 perception on non-pharmaceutical interventions, desire, and behavioral interventions, and as a result of the analysis, COVID-19 perception had a strong negative (−) effect on desire, and desire had a highly positive (+) effect on behavioral intentions. Consequently, the following hypothesis was established.

**Hypothesis 5 (H5).** 
*Desire has a significant effect on behavioral intentions in relation to international travel.*


### 2.6. The Moderation Effect of Gender

Researchers have traditionally placed a strong emphasis on demographic characteristics since they have an impact on behavioral intentions, and gender has long been thought to have an impact on one’s diversified, responsibly pro-sustainable, pro-environmental, and prosocial intentions and behaviors [58] Similarly, the different relationships between gender and health-protective behavioral responses in a pandemic context have been examined through applying the norm-activation model in the study by Abel and Brown [59]. Such different gender expectations are what motivate people to choose different categories of prosocial behaviors. In addition, research on consumer behavior emphasizes gender differences based on social role theory, and when socialization progresses, the roles for males and females are educated as different, and the behavior develops differently [60]. In addition, it was found that males are more active and self-directed than females, especially those who take risks, while females have a greater tendency than males to avoid risks [61].

Based on this context, the research by Bae and Chang [33], which focused on hotel guests, revealed that gender had a significant moderating effect on the connection between affective risk perception and behavioral intention. While emotional risk perception had an equal impact on both genders’ attitudes, it has a bigger negative impact on female participants’ behavioral intentions toward unrestricted travel than it does on male participants. Accordingly, it was intended to understand whether gender moderates the effect of risk perception on psychological perception, desire, and behavioral intention, and to confirm the relationship with the dependent variable, travel intention. Therefore, the following Hypothesis H5 was established.

**Hypothesis 6 (H6a–g).** 
*Gender moderates the relationship between independent variables and behavioral intentions in relation to international travel.*


## 3. Method

### 3.1. Measures and Questionnaire Development

In this study, the MGB from Perugini and Bagozzi [24] is applied to examine the decision-making process for potential travelers’ international travel in dangerous situations such as COVID-19 and war, and specifically, to structurally verify the correlation between risk perception and potential travelers’ behavioral intentions. Furthermore, the Model of Goal-directed Behavior whose validity has been verified in effectively identifying people’s psychology and behavior is applied, and a hypothesis is established based on previous studies [13,34,35,41]. Accordingly, the following model is schematized.

This study aims to verify the correlation between international travel uncertainty and mental well-being for potential travelers in Korea who perceive the risk of COVID-19 and war, and to predict the impact on behavioral intentions based on previous studies applying the Model of Goal-directed Behavior.

The operational definition of the variables used in the analysis is as follows. Risk perception is defined as a basic degree of possible negative consequences to uncertainty of various recognized factors and distinguishing between COVID-19 risk perception and war risk perception, and the details are as follows.

COVID-19 risk perception is defined as the degree of negative response to travel due to the severity of the epidemic caused by COVID-19, and the three questions used in the study by Neuberger and Egger [13] are used. War risk perception is defined as the degree of negative response to travel perceived risk of terrorism, war, etc., and uses the three questions used in the study by Kim, Choi, and Leopkey [35]. Uncertainty about international travel is defined as the degree of uncertainty about economic, psychological, social, and physical losses because destinations are limited internationally due to the risk of COVID-19/war, and five questions used in the study by Chua et al. [34] are used. Mental well-being for international travel is defined as the degree of stress, worry, and anxiety received because travel destinations are limited internationally due to the risk of COVID-19/war, and four questions used in the study by Chua et al. [34] are used. Next, desire and behavioral intention are the variables included in the Model of Goal-directed Behavior, and the questions used in the study by Das and Tiwari [41] are used. Based on this study, desire is defined as a desire for international travel caused by perceiving the risk of COVID-19/war, using 4 questions, and travel intention is defined as the degree to plan international travel by perceiving the risk of COVID-19/war, and 4 questions are used. Demographic questions consist of five questions: gender, age, marital status, education level, and occupation.

### 3.2. Survey Design and Data Collection Process

This study attempts to determine how the COVID-19 risk perception and the war risk perception correlate with the psychological condition and behavior of potential travelers. To this end, five professors majoring in tourism and five researchers at the Ph.D. level or higher were asked to review the questionnaire, and the process of revising and supplementing awkward or unnecessary questions was carried out. In addition, prior to this survey, 20 ordinary people, who were acquaintances of the authors of this study, were asked to respond to the survey, and the final survey was confirmed by reaffirming whether there was any difference from what the expert group understood.

This survey was conducted for one month beginning on 2 September 2022, and a self-written Google Survey Doc was linked to respond on mobile. A survey was conducted on respondents who responded with “yes” to voluntary participation at the beginning of the survey, and only samples of respondents with a level of willingness to respond faithfully to the survey and scored at least 4 points were used for analysis. In addition, Starbucks coupons were presented through a lottery among the respondents who completed the survey to induce an active response. The survey was conducted on potential travelers living in Korea, and specifically, only those who have experienced international travel in the past 1 to 2 years were required to participate in the survey. Convenience sampling was used as a non-probability sampling method, and after 413 copies were distributed, 361 copies were used for the final analysis, excluding 52 copies of unfaithful or ambiguous responses. Frequency analysis, confirmatory factor analysis, discriminant validity analysis, structural model analysis, path analysis, and demographic analysis were conducted, and SPSS and AMOS programs were used as analysis programs. Additionally, verification of the structural relevance between variables was analyzed by covariance-based structural equation modeling (CB-SEM). According to the suggestion of Anderson and Gerbing [62], the suitability verification of the research model was conducted in two stages: the suitability verification of the measurement model and the suitability verification of the structural model. CB-SEM estimates the coefficient using the ML (maximum likelihood) estimation technique so that the difference between the sample covariance matrix and the covariance matrix is minimized.

## 4. Results

### 4.1. Characteristic of Respondents

Of the 361 respondents, 193 respondents (53.46%) were female, and 168 respondents (46.54%) were male. In terms of present marital status, 212 respondents were married (58.73%) followed by 149 respondents (41.27%) who were single. In order from highest to lowest numbers, survey participants were 20–29 years old (125 respondents, 34.63%), 30–39 years old (115 respondents, 31.86%), 40–49 years old (78 respondents, 21.61%), 50 years old (39 respondents, 10.8%), and 60 years old above (4 respondents, 1.11%). As for the level of education, from highest to lowest number of respondents, participants were university students (128 respondents, 35.46%), college students (109 respondents, 30.19%), graduate school students or higher (90 respondents, 24.93%), and high school students (34 respondents, 9.42%); overall, the level of education was high. Job types were salaried employees (163 respondents, 45.15%), students (95 respondents, 26.32%), freelancers (59 respondents, 16.34%), self-employed (37 respondents, 10.25%), and in other types of jobs (7 respondents, 1.94%). The purpose of travel abroad was leisure (319 respondents, 88.37%), business (38 respondents, 10.53%), and other (4 respondents, 1.11%) in order.

### 4.2. Confirmatory Factor Analysis

Before conducting a confirmatory factor analysis (CFA), the data screening was performed to check whether there were any violations of the assumptions. First, the common method bias (CMB) was checked by Harman’s single-factor and common latent factor approaches. All measurement items are loaded into one common factor and the total variance for a single factor was less than 50% (48.23%). There were no significant differences between CFA with and without common latent factor when comparing standardized regression weight. Therefore, CMB may not be caused in our study [63].

The confirmatory factor analysis (CFA) was performed to verify the reliability and validity. As a result of the measurement model, Goodness-of-fit statistics for the measurement model χ^2^ = 313.091, df = 189, *p* < 0.001, χ^2^/df = 1.657, RMSEA = 0.048, GFI = 0.998, CFI = 0.971, IFI = 0.971, TLI = 0.964, were judged to be excellent overall [64]. Factor loadings, significance probability of t−value, average variance extracted (AVE), and construct reliability (CR) were checked to verify the convergent validity of the latent variables of the measurement model. The confidence coefficients (Cronbach’s α) of factor loading were shown between 0.780 and 0.970, which was more significant than the 0.6 suggested by [65]. Moreover, AVE values were constructed ranging from 0.623 to 0.915. These values were all greater than that level of 0.5 and 0.7 suggested by [62].

In addition, correlation analysis was performed as shown in Table 1 to verify discriminant validity. As a result of Pearson’s correlation analysis, all variables of CRP, WRP, UIT, MW, DS, and BI were *p* < 0.05, indicating a significant correlation association [66]. Thus, discriminant validity was confirmed.

### 4.3. Structural Model and Hypothesis Testing

In this study, BI toward international travel was investigated based on the Model of Goal-directed Behavior and the stimulus–organism–response framework. The structural equation model (SEM) analysis was generated by using the maximum likelihood estimation method as an estimation method for both model and procedures’ evaluation [65]. Goodness-of-fit statistics for the structural model: χ^2^ = 355.109 df = 196, *p* < 0.001, χ^2^/df = 1.812, RMSEA = 0.053, GFI = 0.998, CFI = 0.963, IFI = 0.963, TLI = 0.956 was satisfactorily higher than the standard value.

Moreover, SEM had shown high prediction power for R^2^(UIT) = 0.378, R^2^(MW) = 0.370, R^2^(DS) = 0.672, R^2^(BI) = 0.530 and t-values and standardized path coefficient were shown as the result in Table 2. The path estimates show that CRP had a significantly positive effect on UIT (β = 0.050, t = 2.306 *). CRP had a significantly positive effect on MW (β = −0.016, t = −2.369 *). Thus, H1a and H1b were supported. WRP had a significantly positive effect on UIT (β = 0.831, t = 9.555 ***). WRP had a significantly positive effect on MW (β = −0.331, t = −2.851 **). Thus, H2a and H2b were supported. And, UIT had a significantly negative effect on DS (β = −0.208, t = −3.213 ***). Thus, H3 was supported. and, MW had a significantly positive effect on DS (β = 0.650, t = 9.058 ***). Thus, H4 was supported. Lastly, DS had a significantly positive effect on BI (β = 0.108, t = 2.627 **). Thus, H5 was supported.

### 4.4. The Moderation Effect of Gender

To test the five hypotheses that gender moderates the relationships between the five antecedents (CRP, WRP, UIT, MW, DS) and the dependent variable (BI), gender was used to separate the data into two groups (male vs. female). In total, 192 males (53.2%) and 169 females (46.8%) were applied to the observation targets for each group to verify the moderating effect of gender (H6a–g). The baseline models for gender groups were generated by constraining all loadings to be invariant across groups. The results indicated that the baseline model for both gender groups (χ^2^ = 433.209 df = 201, *p* < 0.001, χ^2^/df = 2.155, RMSEA = 0.053, GFI = 0.998, CFI = 0.963, IFI = 0.963, TLI = 0.956) included an acceptable fit to the data. Table 3 and Figure 1 included the results of the invariance tests. The proposed moderating impact of gender (Hypotheses 6a–g) was initially estimated. The baseline model and a series of nested models were compared, where a particular path of interest is restricted to be equal. To identify the difference between groups, a chi-square test was used. Our results of the chi-square test revealed that the paths from CRP → MW(Δχ^2^(1) = 3.883, *p* < 0.05), WRP → MW(Δχ^2^(1) = 4.435, *p* < 0.05) and MW → DS(Δχ^2^(1) = 4.335, *p* < 0.05) differed significantly across gender groups. However, the links from CRP → UIT(Δχ^2^(1) = 1.352, *p* > 0.05), WRP → UIT(Δχ^2^(1) = 3.074, *p* > 0.05), UIT → DS(Δχ^2^(1) = 2.818, *p* > 0.05), DS → BI(Δχ^2^(1) = 2.312, *p* > 0.05) were not significantly different male and female groups. Therefore, while hypotheses 6b,d,f were supported, hypotheses 6a,c,e,g were not supported.

## 5. Discussion

This study tried to investigate the relationship among the behavioral intentions of potential travelers on risk perception. In summary, existing studies [34,47,48] were expanded by further analyzing the risk perception of war beyond the psychological state of travelers regarding COVID-19 risk perception. It shows a similar context as the negative results derived from previous studies on a travel of travelers who were aware of the risk of terrorism [35,49,50], etc. mentioned before COVID-19. In addition, it has academic significance by supporting the results of studies applying the model of goal-oriented behavior [22,56,57]. In particular, the importance of desire is emphasized through its influence on travelers’ behavior.

Although various approaches have been taken in the tourism industry to determine the perception of risk of COVID-19 for travelers [34,44,67,68,69], this study is meaningful by adding war risk perception as an independent variable to specifically analyze travelers’ risk perceptions of COVID-19 and other types of dangerous situations. Moreover, this study examines the correlation between travelers who want to travel internationally at a time when the degree of risk perception may be high using the variables of UIT and MW. We further apply desire, which serves as an important variable in the Model of Goal-directed Behavior, and apply the model based on SOR to identify structural relationships with the variables of the study. In addition, to identify the difference in the influence of each subdivided group based on the characteristics of potential overseas travelers, the moderating effect of gender (male vs. female) was confirmed.

### 5.1. Theoretical and Managerial Implications

There are several academic implications of this study. For Hypothesis 1a, COVID-19 risk perception has a significant positive (+) effect on uncertainty toward international travel, and Hypothesis 1b has a significant negative (+) effect on mental well-being, so Hypothesis 1a,b are adopted. It is shown that war risk perception has a statistically significant positive (+) effect on uncertainty toward international travel, and for Hypothesis 2b, war risk perception has a statistically significant negative (+) effect on mental well-being, so Hypotheses 2a,b are adopted. Hypothesis 3 is adopted as uncertainty toward international travel is found to have a significant negative (−) effect on desire, and Hypothesis 4 is adopted as mental well-being is found to have a significant positive (+) effect on desire. In addition, Hypothesis 5 is adopted as desire as it has a statistically significant positive (+) effect on behavioral intentions toward international travel. Finally, when it comes to the test result of the moderating effect (female vs. male), Hypothesis 6b, d and f have significant effect on moderation. However, Hypothesis 6a, c, e and g have no significant effect on moderation.

This study has evolved from previous studies that predict the behavioral intention of potential travelers due to COVID-19 risk perception. Accordingly, the process applied here explains people’s psychological decision-making processes using various theoretical and logical approaches. The academic implications are as follows. First, understanding the psychological condition of potential travelers and predicting their behavioral intentions through different variables will be important in terms of the perspective of strategic marketing in the tourism industry in the future. In this respect, this study supports the results of previous studies by empirically analyzing perceptions of potential travelers with regard to COVID-19 and war risk [34,35,48,49,50], and contributes to academic understanding by presenting useful analysis to support subsequent studies. Second, since the outbreak of COVID-19 in 2020, most studies in the tourism industry targeting potential travelers have focused only on the risk perception of COVID-19. However, travelers now face a new danger in terms of the Ukrainian–Russian war. Thus, this study presents important evidence for future studies in that few studies about travelers’ behavioral intentions on risk perception of COVID-19 and war currently exist. Third, although empirical analysis was conducted to predict the behavioral intentions of travelers in previous studies by applying the Model of Goal-directed Behavior against the backdrop of COVID-19, this study verifies the correlation between respondents’ DS and BI by simultaneously considering the risk situation of COVID-19 and war and presents additional basic evidence for research to apply the Model of Goal-directed Behavior in the future. Fourth, as a leading study, this study presented a research model that predicts potential travelers’ future overseas travel behavior during the COVID-19 pandemic and war risk, and it can be used as comparative data for future study of gender behavior according to risk perception.

In circumstances such as the COVID-19 pandemic and war, it is difficult to control for their influence; therefore, efforts should be made to carry out a long-term and large-scale study. In this respect, the following practical implications are provided through this study. First, a stable policy plan for the tourism industry between each country should be prepared. If the COVID-19 prevention and control measures and the safety zone from war and terrorism are made clear and thorough, travelers who want to travel for leisure and business will be able to plan in advance in a safe psychological state. These results are similar to the practical implications of a study by Zhang et al. [69] that verified a statistically significant correlation between travel behavior before and during COVID-19 for Hong Kong travelers. Second, the degree of uncertainty toward international travel and mental well-being toward international travel has a statistically significant effect on the desire of travelers planning international travel. From the perspective of enterprise, tourism companies should establish a thorough plan to ensure that travelers are efficient with regard to a series of processes, such as the quantity and quality of information before travel, the consistency and usefulness of information during travel, and post-travel satisfaction. It is a way to ensure that the level of information provided by travel companies is consistent with the destination to visit and that embassies in each country can protect their citizens through medical and health support. Third, to help travelers feel safe in planning travel to countries around the world, companies related to well-known destinations need various activities that can instill an image to ensure safety from risk for local residents and visitors, not just an image to generate profits. In addition, as verified by many previous studies on the image of tourist destinations, social and psychological safety should be emphasized to revitalize existing tourist destinations.

### 5.2. Limitations and Future Suggestions

First, there was a limit to the representativeness of the sample, which comprised a significantly small number of respondents in their 50s or older. In dangerous situations such as the COVID-19 pandemic and the Ukrainian–Russian war, younger generations have attempted to plan international travel. It can be assumed that those in their 50s and older are not within the scope of respondents who have experienced international travel because they have underlying diseases and may have great concerns and fears about it.

Second, the degree of risk perception, desire, and behavioral intention may vary depending on the destination of international travel, and this study focuses on international travel more generally. For example, the image of safe quarantine and COVID-19 may differ depending on destination. Therefore, in future studies, it will be necessary to select more specific targets within the scope of general international travel. Thus, a more meaningful study would be possible by investigating the most popular tourist destinations of Koreans into China and the United States for comparison and verification.

Third, in the case of COVID-19, since it has not yet ended, there may be either a direct or indirect effect on the degree of risk perception for respondents. In other words, if the respondent or a close acquaintance is infected, they may respond very negatively to international travel. In future studies, it would be better to ask whether respondents have experienced COVID-19 infection and analyze the behavioral intention of international travel by distinguishing infected and non-infected people.

Fourth, the moderating effect (H6b,d,f) as a significant result found that women had a higher risk perception than men. Both women and men showed no difference in the influence of CRP and WRP on UIT, but women perceived both CRP and WRP as higher than men. In addition, in terms of influence on DS, there was no difference between men and women in UIT, but MW was higher in women than in men. This may be a part of South Korea’s somewhat flexible acceptance of the fear or fear of men’s perception of war due to the nature of the country, which is still in a truce after the war with North Korea. However, since women still have a significantly higher influence on MW than men, the purchasing power in the travel market is largely determined by women, so it is necessary to properly utilize the package composition that can stabilize their MW, reliability of guide education on travel safety, cleanliness, and stability of accommodation areas.

## Figures and Tables

**Figure 1 ijerph-20-02562-f001:**
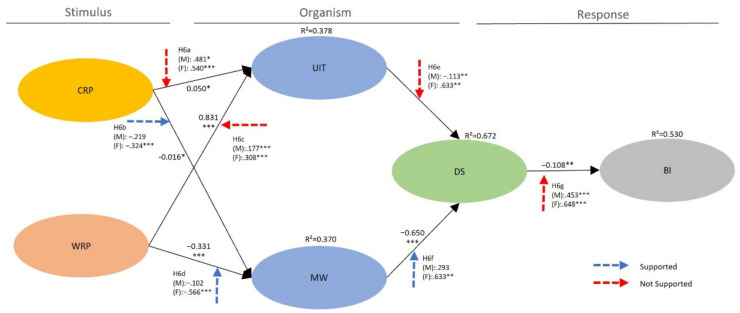
Structural equation model estimation and test for structural metric invariance. Note 1. COVID-19 risk perception (CRP), war risk perception (WRP), uncertainty toward international travel (UIT), mental well-being toward international travel (MW), desire (DS), behavioral intentions toward international travel (BI). Note 2. Goodness-of-fit statistics for the measurement model χ^2^ = 313.091, df = 189, *p* < 0.001, χ^2^/df = 1.657, RMSEA = 0.048, GFI = 0.998, CFI = 0.971, IFI = 0.971, TLI = 0.964. Note 3. Goodness-of-fit statistics for the structural model: χ^2^ = 355.109 df = 196, *p* < 0.001, χ^2^/df = 1.812, RMSEA = 0.053, GFI = 0.998, CFI = 0.963, IFI = 0.963, TLI = 0.956; Note 4. Goodness-of-fit statistics for the baseline model: χ^2^ = 433.209 df = 201, *p* < 0.001, χ^2^/df = 2.155, RMSEA = 0.053, GFI = 0.998, CFI = 0.963, IFI = 0.963, TLI = 0.956; * *p* < 0.05, ** *p* < 0.01, *** *p* < 0.001.

**Table 1 ijerph-20-02562-t001:** The measurement model and correlation.

Construct and Scale Item		Standardized Loading	Mean (SD)	AVE (CR)	CRP	WRP	UIT	MW	DS	BI	√AVE
CRP	CRP1	0.952	3.819 (0.622)	0.915 (0.970)	-						0.957
	CRP2	0.948									
	CRP3	0.970									
WRP	WRP1	0.894	4.142 (0.360)	0.752 (0.898)	0.002 *	-					0.867
	WRP2	0.906									
	WRP3	0.804									
UIT	UIT1	0.794	4.092 (0.493)	0.699 (0.881)	−0.071 **	−0.527 ***	-				0.836
	UIT2	0.826									
	UIT3	0.835									
	UIT4	0.683									
	UIT5	0.674									
MW	MW1	0.755	3.925 (0.572)	0.664 (0.804)	−0.044 *	−0.414 ***	0.508 ***	-			0.815
	MW2	0.714									
	MW3	0.622									
	MW4	0.561									
DS	DS1	0.831	1.950 (0.479)	0.623 (0.837)	−0.081 **	−0.362 ***	−0.539 ***	−0.611 ***	-		0.789
	DS2	0.735									
	DS3	0.779									
BI	BI1	0.503	2.383 (0.394)	0.684 (0.780)	−0.023 ***	−0.205 ***	−0.136 ***	−0.118 *	−0.100 **	-	0.827
	BI2	0.782									
	BI3	0.834									
	BI4	0.625									

Note 1. SD = standardized deviation, AVE = average variance extracted, CR = composite reliability, COVID-19 risk perception (CRP), war risk perception (WRP), uncertainty toward international travel (UIT), mental well-being toward international travel (MW), desire (DS), behavioral intentions toward international travel (BI). Note 2. Goodness-of-fit statistics for the measurement model χ^2^ = 313.091, df = 189, *p* < 0.001, χ^2^/df = 1.657, RMSEA = 0.048, GFI = 0.998, CFI = 0.971, IFI = 0.971, TLI = 0.964, * *p* < 0.05, ** *p* < 0.01, *** *p* < 0.001.

**Table 2 ijerph-20-02562-t002:** Structural model results and hypothesis testing.

Hypothesized Paths	Coefficients	Std. Error	t-Values
H1a: CRP → UIT	0.050	0.038	2.306 *
H1b: CRP → MW	−0.016	0.044	−2.369 *
H2a: WRP → UIT	0.831	0.087	9.555 ***
H2b: WRP → MW	−0.331	0.116	−2.851 **
H3: UIT → DS	−0.208	0.065	−3.213 ***
H4: MW → DS	0.650	0.072	9.058 ***
H5: DS → BI	0.108	0.041	2.627 **
Explained variable:	R^2^(CRP) = 0.378, R^2^(WRP) = 0.370, R^2^(DS) = 0.672, R^2^(BI) = 0.530		

Note 1. COVID-19 risk perception (CRP), war risk perception (WRP), uncertainty toward international travel (UIT), mental well-being toward international travel (MW), desire (DS), behavioral intentions toward international travel (BI). Note 2. Goodness-of-fit statistics for the structural model: χ^2^ = 355.109 df = 196, *p* < 0.001, χ^2^/df = 1.812, RMSEA = 0.053, GFI = 0.998, CFI = 0.963, IFI = 0.963, TLI = 0.956, * *p* < 0.05, ** *p* < 0.01, *** *p* < 0.001.

**Table 3 ijerph-20-02562-t003:** Results of the moderating effect of gender.

Paths.	Male (N = 193)		Female (N = 168)		Baseline Model (Freely Estimated)	Nested Model (Constrained to Be Equal)
	Coefficients	t-Value	Coefficients	t-Value		
H6a: CRP → UIT	0.481	2.394 *	0.540	3.231 ***	χ^2^(201) = 433.209	χ^2^(202) = 434.561 ^a^
H6b: CRP → MW	−0.219	−0.834	−0.324	−3.829 ***	χ^2^(201) = 433.209	χ^2^(202) = 437.092 ^b^
H6c: WRP → UIT	0.177	3.310 ***	0.308	4.234 ***	χ^2^(201) = 433.209	χ^2^(202) = 436.283 ^c^
H6d: WRP → MW	−0.102	−0.029	−0.566	−4.288 ***	χ^2^(201) = 433.209	χ^2^(202) = 437.644 ^d^
H6e: UIT → DS	−0.113	−2.394 **	−0.192	−3.922 ***	χ^2^(201) = 433.209	χ^2^(202) = 436.027 ^e^
H6f: MW → DS	0.293	1.029	0.633	6.771 **	χ^2^(201) = 433.209	χ^2^(202) = 437.544 ^f^
H6g: DS → BI	0.453	3.323 ***	0.648	3.016 ***	χ^2^(201) = 433.209	χ^2^(202) = 435.521 ^g^
Chi-square difference test:			Test results:		Goodness-of-fit statistics for the baseline model: χ^2^ = 433.209 df = 201, *p* < 0.001, χ^2^/df = 2.155, RMSEA = 0.043, GFI = 0.968, CFI = 0.933, IFI = 0.913, TLI = 0.926 * *p* < 0.05, ** *p* < 0.01, *** *p* < 0.001	
aΔχ^2^(1) = 1.352, *p* > 0.05			H6a: Not supported			
bΔχ^2^(1) = 3.883, *p* < 0.05			H6b: Supported			
cΔχ^2^(1) = 3.074, *p* > 0.05			H6c: Not supported			
dΔχ^2^(1) = 4.435, *p* < 0.05			H6d: Supported			
eΔχ^2^(1) = 2.818, *p* > 0.05			H6e: Not supported			
fΔχ^2^(1) = 4.335, *p* < 0.05			H6f: Supported			
gΔχ^2^(1) = 2.312, *p* > 0.05			H6g: Not supported			

Note 1. COVID-19 risk perception (CRP), war risk perception (WRP), uncertainty toward international travel (UIT), mental well-being toward international travel (MW), desire (DS), behavioral intentions toward international travel (BI).

## Data Availability

The data presented in this study are available on request from the corresponding author. The data are not publicly available due to confidentiality agreements with participants.

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
