# Peer review of "Applying a Goal-Directed Behavior Model to Determine Risk Perception of COVID-19 and War on Potential Travelers’ Behavioral Intentions"

_ijerph, 2023, doi:10.3390/ijerph20032562_

Round 1
Reviewer 1 Report
The study is very interesting and timely. The theoretical review and methodology are very appropriate. However, I believe that certain aspects should be modified prior to publication.
1. The results of the abstract should be rewritten. I see no point in listing which hypotheses are accepted and which are rejected. It is important to summarize and highlight the most interesting results.
2. Regarding the discussion, the excessive use of abbreviations to name the variables of the study makes it difficult to read. The same occurs with the Theoretical and Managerial Implications. A table of abbreviations should be included and even in the discussion and implications sections we should try to avoid their use.
3. Regarding the Theoretical and Managerial Implications, I think that the first paragraph does not make sense. It should be reworded or deleted. In this paragraph the results of the study are repeated with excessive use of abbreviations. I think this is not correct and the reading is tedious.
Moreover, in my opinion, the theoretical implications of the study should be more clearly highlighted.
4. In my opinion, a figure of the proposed theoretical model would enrich and clarify the study.
Author Response
Dear Editor and reviewers,
Thank you very much for reviewing our manuscript. We appreciate the opportunity you have afforded us to revise and resubmit. We found your suggestions to be thought-provoking and useful and have worked diligently to improve the manuscript as you suggested. Below, you will find our replies and responses to your constructive comments. Within the responses, red sections denote changes we made to the manuscript itself. Within the manuscript itself, changes are highlighted in red. We hope the changes made are satisfactory to you.
************************************************************************************
Point 1: The results of the abstract should be rewritten. I see no point in listing which hypotheses are accepted and which are rejected. It is important to summarize and highlight the most interesting results.
Response 1: Thank you for your comment. We have corrected the part pointed out by the reviewer correctly.
Point 2: Regarding the discussion, the excessive use of abbreviations to name the variables of the study makes it difficult to read. The same occurs with the Theoretical and Managerial Implications. A table of abbreviations should be included and even in the discussion and implications sections we should try to avoid their use.
Response 2: Thank you for your comment. We have corrected the part pointed out by the reviewer correctly.
Point 3: Regarding the Theoretical and Managerial Implications, I think that the first paragraph does not make sense. It should be reworded or deleted. In this paragraph the results of the study are repeated with excessive use of abbreviations. I think this is not correct and the reading is tedious.
Moreover, in my opinion, the theoretical implications of the study should be more clearly highlighted.
Response 3: Thank you for your comment and delete the first paragraph. We have corrected the part pointed out by the reviewer correctly.
Point 4: In my opinion, a figure of the proposed theoretical model would enrich and clarify the study.
Response 4: Thank you for your comment. We have submitted the figure of the study.

Reviewer 2 Report
I found this paper well written after a careful review. A few issues for further improvement are below:.
1. The research model involving hypotheses should be represented graphically, while the Figure1 has not been properly inserted in the paper.
2. The paper is about applying a goal-Directed Behavior Model to Determine Risk Perception of COVID-19 and War on Potential Travelers’ Behavioral Intentions, however the associations among travel risk perception under post-epidemic era, global motives and behavioral intentions have not been thoroughly described in both Introduction and Literature Review Parts, need to be majorly revised. This paper can be further enriched by citing more related previous research. Below are some of the papers authors consider to improve the quality of paper.
Chi, X., Meng, B., Lee, H., Chua, B.-L., & Han, H. (2023). Pro-environmental employees and sustainable hospitality and tourism businesses: Exploring strategic reasons and global motives for green behaviors. Business Strategy and the Environment, 1–16. https://doi.org/ 10.1002/bse.3359
Chen, H., Wang, L., Xu, S., Law, R., & Zhang, M. (2023). Research on the Influence Mechanism of Intention to Proximity Travel under the COVID-19. Behavioral Sciences, 13(1), 10.
Billore, S., Anisimova, T., & Vrontis, D. (2023). Self-regulation and goal-directed behavior: A systematic literature review, public policy recommendations, and research agenda. Journal of Business Research, 156, 113435.
3. Theoretical and managerial discussions are strongly suggested to be separated into two parts in a logical way, and the depth of discussions need to be strengthened through comparing the previous literatures and exploring more effective managerial implications.
Author Response
Dear Editor and reviewers,
Thank you very much for reviewing our manuscript. We appreciate the opportunity you have afforded us to revise and resubmit. We found your suggestions to be thought-provoking and useful and have worked diligently to improve the manuscript as you suggested. Below, you will find our replies and responses to your constructive comments. Within the responses, red sections denote changes we made to the manuscript itself. Within the manuscript itself, changes are highlighted in red. We hope the changes made are satisfactory to you.
************************************************************************************
Point 1: The research model involving hypotheses should be represented graphically, while the Figure1 has not been properly inserted in the paper.
Response 1: Thank you for your comment. We have submitted the figure of the study hypothesis to editor and will correct more precisely soon.
Point 2: The paper is about applying a goal-Directed Behavior Model to Determine Risk Perception of COVID-19 and War on Potential Travelers’ Behavioral Intentions, however the associations among travel risk perception under post-epidemic era, global motives and behavioral intentions have not been thoroughly described in both Introduction and Literature Review Parts, need to be majorly revised. This paper can be further enriched by citing more related previous research. Below are some of the papers authors consider to improve the quality of paper.
Response 2: Thank you for your comment and the article (Chen et al., 2023; Chen et al., 2023). We have added and revised the part pointed out by the reviewer correctly.
Point 3: Theoretical and managerial discussions are strongly suggested to be separated into two parts in a logical way, and the depth of discussions need to be strengthened through comparing the previous literatures and exploring more effective managerial implications.
Response 3: Thank you for your comment and divided two parts theoretical and managerial. We have revised and corrected the part pointed out by the reviewer correctly.

Reviewer 3 Report
Thank you for giving me the opportunity to review the manuscript. The subject is actual and the manuscript is giving answers for the practitioners in tourism area. the manuscript is well structured and presented.
Some recommendations:
1- because there are 6 H with sub-hypothesis a table with a synthesis of the results will better clarify the findings.
2- highlight the convergence/divergence between Covid-19 and war if possible. It is not clear if the challenges were consider bulk or there are data to compare.
3- highlight and explain if the results are applicable to any kind of challenge or there are some characteristics of the event to be considered.
Author Response
Dear Editor and reviewers,
Thank you very much for reviewing our manuscript. We appreciate the opportunity you have afforded us to revise and resubmit. We found your suggestions to be thought-provoking and useful and have worked diligently to improve the manuscript as you suggested. Below, you will find our replies and responses to your constructive comments. Within the responses, red sections denote changes we made to the manuscript itself. Within the manuscript itself, changes are highlighted in red. We hope the changes made are satisfactory to you.
************************************************************************************
Point 1: because there are 6 H with sub-hypothesis a table with a synthesis of the results will better clarify the findings.
Response 1: Thank you for your comment. We have corrected the part pointed out by the reviewer.
Point 2: highlight the convergence/divergence between Covid-19 and war if possible. It is not clear if the challenges were considering bulk or there are data to compare.
Response 2: Thank you for your comment. Those who perceived the risk of COVID-19 and war tried to investigate the difference in psychological changes of travelers toward overseas travel. The theoretical background and analysis results were also aimed at verifying the relationship. We will refer to the opinions by reviewer and conduct an empirical analysis in consideration of this in future studies.
Point 3: highlight and explain if the results are applicable to any kind of challenge or there are some characteristics of the event to be considered.
Response 3: Thank you for your comment. We have corrected and revised the part pointed out by the reviewer.
